# Evaluation of the Reliability of Calibration Constants in Heat Flux Meters Using an ISO 5660-1 Cone Heater

**DOI:** 10.3390/s25247406

**Published:** 2025-12-05

**Authors:** Woo-Geun Kim, Cheol-Hong Hwang, Sung Chan Kim

**Affiliations:** 1Department of Fire and Disaster Prevention, Daejeon University, 62 Daehak-ro, Dong-gu, Daejeon 34520, Republic of Korea; fayageun@gmail.com; 2School of Fire and Disaster Management, Kyungil University, 33 Buho-ri, Hayang-eup, Gyungsan 38428, Republic of Korea; sungkim@kiu.ac.kr

**Keywords:** heat flux meter, secondary calibration, calibration constant, cone calorimeter

## Abstract

This study evaluates the reliability of calibration constants for heat flux meters (HFMs) using a secondary standard under controlled thermal conditions provided by an ISO 5660-1 cone heater. Six HFMs (three new and three previously used) were examined by comparing manufacturer-provided (MFG) and secondary-standard-based (SEC) calibration constants. The bias between the MFG and SEC calibration constants varied substantially depending on the manufacturer, calibration method, service history, and the coating condition of the sensing surface. When the Bland–Altman limits of agreement were defined with the new HFMs as the reference, an HFM from a different manufacturer or calibrated by a different method was found to fall outside these limits. Although the absolute accuracy of the secondary-standard HFM has not been independently validated, the approach using the cone heater is practical for field implementation in terms of equipment accessibility, cost, and operational simplicity. Recalibration against the same secondary-standard prior to use is recommended to ensure repeatability and reproducibility across experiments and institutions.

## 1. Introduction

Heat flux, along with temperature, is a fundamental physical quantity for quantifying the thermal behavior of fires. Heat flux meters (HFMs) are widely used in various applications, including full-scale fire experiments, combustion property assessments, and evaluations of fire-resistive material performance, and they serve as indispensable measurement instruments for fire dynamics analysis and performance-based design (PBD) [1,2,3]. In practice, HFMs are typically supplied with manufacturer-provided (MFG) calibration constants or calibration equations, which are often used directly in experiments without independent verification. However, such constants inherently contain uncertainties arising from manufacturing conditions, calibration environments, measurement instrumentation, and surface properties. Accordingly, even new HFMs require experimental validation by the user to ensure reliability. Moreover, repeated use of HFMs in high-temperature environments can lead to changes in sensitivity due to coating degradation, contamination, or material deterioration of the sensing surface, further reducing the validity of the initial calibration constants over time [4,5]. Consequently, systematic verification of the initial calibration constants and periodic reassessment considering service history are essential. Establishing standardized verification and recalibration procedures is critical to ensuring the long-term measurement reliability of HFMs [6,7].

Maintaining the reliability of HFMs requires periodic calibration and consideration of sensitivity changes during operational lifetime. Calibration methods are generally classified into primary calibration, which employs high-precision blackbody radiators or standard optical sources, and secondary calibration, which compares the test sensor with a reliable reference sensor [6,8,9,10,11,12]. However, most laboratories lack the costly facilities required for primary calibration, resulting in a strong demand for secondary calibration procedures with high accessibility and repeatability [13]. When properly maintained reference sensors and stable heat sources are used, secondary calibration can provide comparatively high accuracy even under limited laboratory resources. For these reasons, assessing the applicability of practical secondary-standard calibration procedures that researchers can implement directly is an important step toward ensuring the reliability of measurements in fire research and related testing.

Calibration methods and devices for HFMs vary significantly across research institutions, with a wide range of approaches applied to both primary and secondary calibration. In addition to the selective application of ISO standards related to primary and secondary calibration, each laboratory follows its internal standard procedures. A representative study conducted by Pitts et al. [6] involved an international round-robin test in which several authoritative research institutions applied different equipment and procedures to calibrate HFMs. The results showed generally acceptable levels of agreement for most fire research and standard testing purposes. However, when examined in detail, differences in the calibration constants were observed depending on the heat source characteristics, calibration methods, and sensor types, with deviations typically within ±5%. In some cases, deviations were observed to extend up to approximately 10%. More recently, Kim and Hamins [14] recalibrated three Schmidt–Boelter-type HFMs with no prior usage history using the NIST’s lamp-based calibration system. Their results showed relative errors in the MFG calibration constants ranging from 1.5% to a maximum of 14.3%. This finding suggests that even when sensors are in new condition, significant uncertainties can exist in the MFG calibration constants, underscoring the necessity for researchers to independently perform calibration procedures and verify these constants.

To validate the calibration constants of a new HFM and reassess its calibration constants based on practical usage, it is essential to provide a method that is easily accessible to researchers. Fortunately, the ISO 5660-1 apparatus [15,16] for combustion property measurements is available in most laboratories, and the cone heater integrated into this system can be effectively utilized for heat flux calibration [17]. The ISO 5660-1 apparatus is fundamental in replicating realistic fire conditions, enabling accurate measurement of the thermal characteristics observed during fire testing, which are crucial for fire safety assessments and performance-based design (PBD) [18]. In addition to measuring heat release rate (HRR), this apparatus is pivotal in simulating diverse fire scenarios.

The cone heater in the ISO 5660-1 apparatus not only allows precise regulation of heat flux but is also widely adopted, offering substantial practical advantages. Specifically, the cone heater is designed to provide a relatively uniform spatial distribution of heat flux across a 100 mm × 100 mm specimen [15], thereby enabling precise control of heat flux during experiments. Although the cone heater does not provide a perfectly uniform heat flux across the entire 100 mm × 100 mm specimen [19,20,21,22], it is anticipated that adequate calibration can be achieved within an acceptable error range when the specimen is removed and an HFM with a relatively small sensing area is centered at the specimen location, with its sensing surface aligned with the plane of the original specimen surface. Furthermore, the cone heater creates a radiative environment that closely mimics real fire conditions, offering significant potential as a calibration platform in terms of operational repeatability and heat flux stability. Despite these advantages, the methodologies for calibrating HFMs using the ISO 5660-1 apparatus have not yet been comprehensively standardized, and quantitative data regarding the variation in calibration constants due to different sensor types and operating conditions remain limited.

Against this background, a practical secondary-standard calibration procedure employing an ISO 5660-1 cone heater as the radiative source was implemented, and calibration constants for six HFMs—three new and three previously used—were experimentally determined and verified. Systematic bias and deviations under realistic operating conditions were quantified through comparison with MFG constants, and the effects of surface degradation and recoating on sensitivity and the resulting calibration constants were evaluated. Additionally, a Bland–Altman analysis was used to estimate bias and the limits of agreement (LOA). Based on these results, the validity and limitations of heat flux calibration against a secondary standard were discussed. The combined experimental and statistical results suggested the practical potential of ISO 5660-1–based calibration as an alternative in terms of repeatability and accessibility. By presenting quantitative comparative data across meters with different service histories, groundwork for laboratory-level calibration procedures was provided, and the reliability of heat flux measurements in fire experiments is expected to be enhanced.

## 2. Experimental Conditions

### 2.1. Experimental Method and Procedures

The calibration of the HFMs was carried out by a secondary-standard calibration procedure in which the cone heater of the ISO 5660-1 apparatus served as the radiative source. A secondary-standard HFM, calibrated using the lamp-based calibration system developed by the NIST Fire Research Division, was obtained and used as the reference HFM [14]. The NIST calibration source consists of a lamp and an ellipsoidal reflector and employs a 2000 W tungsten–halogen filament lamp. Its radiation passes through a 6.4 cm × 6.4 cm square aperture before reaching the sensor position, and the lamp output is controlled by a DC power supply, while the HFM signal is measured with a digital voltmeter. The calibration constant of this reference HFM was applied to determine the reference heat flux at each prescribed heat flux level, and the calibration constants of the test HFMs (three new and three previously used) were subsequently derived with respect to this reference.

Figure 1 presents a schematic of the ISO 5660-1 apparatus [15] on which the calibration was performed. The cone heater and exhaust hood were operated, and the sensing surface of the HFMs was positioned at the specimen top-surface reference level, which is 25 ± 1 mm below the lowest level of the cone heater, as specified in ISO 5660-1. To ensure repeatable alignment and positional reproducibility, a dedicated bracket was fabricated to replace the specimen holder, allowing both 0.5-inch (small) and 1.0-inch (large) HFMs to be mounted concentrically on the cone axis. A push-fit connection (see the inset of Figure 1) was incorporated so that the elevation of the sensing surface was set reproducibly without operator dependence. The exhaust hood was operated at 0.024 m^3^/s in accordance with the standard to suppress thermal-flow accumulation and recirculation near the cone and thereby enhance measurement repeatability. Under these conditions, a stable environment suitable for the calibration of total heat flux (including both radiative and convective components) was established.

The calibration of the HFMs was conducted in the following sequence. Power was supplied to the cone heater and controlled to reach the target output, after which the system was maintained for approximately 20 min to achieve quasi-steady conditions. The output voltage (mV) of the secondary-standard HFM was subsequently recorded for 10 min and time-averaged, after which its calibration constant was applied to obtain the reference heat flux. For each HFM and each prescribed heat flux level, the reference heat flux was first determined and verified using the secondary-standard HFM, and the calibration of the test HFMs was then performed sequentially with respect to this reference. The prescribed heat flux levels were 3, 6, 9, 12, 15, and 20 kW/m^2^, all within the maximum calibration range (20 kW/m^2^) of the secondary standard. Under quasi-steady conditions, the relative fluctuation of the output signal was found to be 0.33–0.53% across the full range of heat flux settings, confirming that the time-averaged signal was an appropriate representative value. Three repetitions were performed for all conditions, and the relative standard deviation for repeatability is presented as vertical error bars in the results. Cooling water was supplied to the HFMs at 0.7 L/min, and the water temperature was maintained within 10–30 °C as recommended by the manufacturer. In this study, the inlet temperature was measured at 20.3 ± 0.6 °C. It is noted that variations in cooling-water temperature can affect the absolute heat flux through changes in the regression intercept (y-intercept), but they do not have a meaningful effect on the slope (calibration constant) relating output voltage to heat flux [14].

The appropriateness of placing the HFMs on the cone axis was verified by a spatial uniformity assessment of the external heat flux, as shown in Figure 2. All HFMs used in this study had a sensing-surface diameter of 1.0 cm. ISO 5660-1 specifies ±2% uniformity within a 2.5 cm radius at 50 kW/m^2^. To verify this requirement, a dedicated radial-scanning mounting fixture described in a previous study [21] was used, and radial external heat flux distributions were measured at 10, 25, and 50 kW/m^2^. At the specimen top-surface reference level, the external heat flux was observed to decrease gradually with increasing distance from the cone axis. Nevertheless, within a radial distance of up to 3.5 cm from the center, the deviation from the central value remained within ±2%, thereby fully satisfying the uniformity requirement. Therefore, when the HFMs are positioned at the specimen top-surface reference level specified by ISO 5660-1 and mounted on the cone axis, calibration can be performed in a uniform radiative environment with precision and reproducibility.

### 2.2. Specifications and Conditions of HFMs Examined

Table 1 summarizes, for the HFMs examined, the photographs, condition (new/used), outer diameter, sensor type, maximum capacity, years in service (used only), and the MFG calibration constants. The notation is N (new) and U (used); size classes are S (small, 0.5 in; cone-calorimeter type) and L (large, 1.0 in; full-scale fire-testing type). This tabulation facilitates direct comparison of specifications and service histories and provides a reference for the interpretation of the calibration results presented in the subsequent sections. For clarity of interpretation, a manufacturer (code) column is included in Table 1, and the actual names are anonymized as A and B. The standards underlying the calibration methodologies are specified differently by the manufacturers. Manufacturer A applied ISO 14934-3 [7], whereas Manufacturer B followed ISO/IEC 17025 [23], ANSI/NCSL Z540-1 [24], and MIL-STD-45662A [25].

The new HFMs (N-S1, N-L1, and N-L2) were included as reference cases for a quantitative assessment of the validity of the MFG calibration constants. The used HFMs (U-S1, U-L1, and U-L2) were selected to reflect differences in surface-coating condition and years in service (including partial coating damage, no visible damage, and complete coating removal). Under an identical secondary-standard calibration procedure, deviations from the MFG calibration constants were quantified so that the range of measurement error attributable to service history and surface condition could be systematically examined. For U-L2, a pair of states was considered: U-L2a (coating completely removed) and U-L2b (recoated). Recoating was performed in the laboratory using HiE-COAT^TM^ 840-CM (Aremco Product, Inc., Valley Cottage, NY, USA; specified emissivity ε = 0.9). This pairing enabled evaluation of the calibration feasibility under an extreme degradation condition and of the extent to which measurement reliability can be restored by recoating.

## 3. Results and Discussion

In this section, the reliability of calibration constants for HFMs, including new and used instruments, was evaluated using a secondary-standard-based (SEC) calibration with an ISO 5660-1 cone heater as the heat source. The slope (calibration constant) and the y-intercept (offset) obtained from the MFG and SEC calibration equations were first compared quantitatively, and differences between the two calibrations (MFG and SEC) were identified. The relative deviation of heat flux values calculated using the MFG calibration relative to the reference heat flux was then assessed. The effect of maintenance was examined by comparing a degraded sensor before and after recoating. Finally, a Bland–Altman analysis was performed to estimate the bias and the limits of agreement (LOA), and the resulting estimates were interpreted relative to secondary-standard calibration values.

Figure 3 presents, based on the sensor configuration in Table 1, the relationship between measured voltage (mV) and the reference heat flux (kW/m^2^) for new and used HFMs after calibration using a secondary-standard HFM. Because the calibration was performed with reference to the secondary standard, the y-axis is labeled as reference heat flux. The linear voltage–heat flux relationship observed here is consistent with a thermoelectric response governed by radiative heat absorption at the sensing surface and subsequent heat conduction to the thermopile junctions. Under these conditions, the slope represents an effective sensitivity determined by emissivity and cooling, whereas any intercept primarily reflects setup dependent offsets. The evaluation was conducted within the calibration range of the secondary-standard HFM (20 kW/m^2^), over which all HFMs exhibited very high linearity. The new HFMs in Figure 3a show similar slopes and y-intercepts close to zero, indicating consistent sensitivity and a small offset in the low heat flux region. In contrast, the used HFMs in Figure 3b display a wider spread in slope and y-intercept, with some positive or negative intercepts. In particular, U-L2a, whose protective coating was fully damaged (Table 1), shows the lowest linearity in the group (R^2^ = 0.9987) and a negative y-intercept, indicating that complete coating loss can affect sensitivity (slope) and offset (y-intercept). Note that the slopes and intercepts in this figure were obtained from SEC calibration equations, so performance should not be judged solely from absolute values across different sensors.

Figure 4 compares the MFG calibration constants with the SEC calibration constants for new and used HFMs. Parenthetical values above each bar denote the relative error of MFG with respect to SEC, expressed as a percentage (negative indicates MFG < SEC; positive indicates MFG > SEC). As shown in Figure 4a, for the new HFMs, SEC exceeded MFG in all cases, with relative errors of −17.9%, −14.5% and −15.8%, respectively, indicating a systematic negative bias. Given that all three HFMs are from the same manufacturer, this consistency indicates that the MFG calibration constants are systematically lower than those derived from the secondary standard. In contrast, the used HFMs (Figure 4b) exhibited substantial unit-to-unit variability. U-S1 (partial coating damage) yielded SEC < MFG (+15.4%), whereas U-L1 (no visible coating damage) showed −9.6%, i.e., the same sign as observed for the new-HFM group. Notably, U-L1 is the only used HFM from a different manufacturer. As summarized in Table 1, inter-manufacturer differences in calibration methodology likely account for its opposite trend relative to the other used HFMs. U-L2a (coating removed) exhibited the largest deviation (−40.8%), consistent with reduced responsivity due to loss of the high-emissivity coating and the consequent requirement for a larger SEC calibration constant. Recoating the same HFM with a nominal emissivity of 0.9 (U-L2b) substantially reduced the discrepancy to −10.9%, bringing it close to the manufacturer value.

Table 2 summarizes, for each HFM, the linear calibration parameters (slope and y-intercept) for MFG and SEC, enabling quantitative comparison with Figure 3 and Figure 4. The listed uncertainties were obtained from linear-regression fits to the calibration data pooled over three repeated runs (*n* = 3) and are reported as two standard deviations (2σ) of the regression estimate. These values represent Type-A repeatability only and should not be interpreted as combined or expanded calibration uncertainties [6]. In some HFMs, the SEC y-intercept was larger than the MFG value; this can be attributed to differences in calibration conditions, particularly the coolant-water temperature. As noted in Section 2, changes in coolant-water temperature can shift the y-intercept and affect the absolute heat flux determination, whereas the slope is not significantly affected [14]. Additional contributors to differences in y-intercept may include background heat flux and shielding conditions, amplifier offset and zero setting, and the assumed emissivity and condition of the coating [8,10,26].

Figure 5 compares the heat flux converted from the mV outputs of the new HFMs using the calibration equations provided by the manufacturer with the reference values determined by a secondary-standard HFM. It also presents the heat flux dependence of the relative error. In Figure 5a, all heat flux values determined using the MFG calibration constant lie below the identity line (y = x). The MFG calibration constants are smaller than the references, so the converted heat flux values are systematically underestimated to approximately 84% of the references across the entire range. As also observed in Figure 3 and Figure 4, the similarity of calibration constants among the HFMs yields similar trends in the converted heat fluxes as a function of the reference heat fluxes. In Figure 5b, the absolute value of the relative error decreases with increasing reference heat flux and then gradually levels off. This behavior occurs because the offset that is prominent at low heat flux becomes less influential as the flux increases. The remaining error approaches a constant level governed primarily by gain mismatch. Quantitatively, the unit with the largest deviation, N-S1, decreases from about −22% at 3 kW/m^2^ to −19% at 20 kW/m^2^. The unit with the smallest deviation, N-L1, ranges from approximately −18% to −15% over the same interval. Therefore, even for new HFMs, direct application of the MFG calibration equation requires considerable caution. To secure consistent experimental results and enable reliable comparisons across studies, recalibration against the same secondary-standard HFM must be performed prior to use.

Figure 6 presents results for the used HFMs (U-S1, U-L1, and U-L2a). It compares the heat flux converted using the MFG calibration equation with the reference heat flux determined by a secondary-standard HFM, and reports the relative error with respect to the reference. Figure 6a presents a direct comparison between the heat flux converted using the MFG calibration equation and the reference heat flux. U-S1 is distributed predominantly above the identity line (y = x), indicating overestimation relative to the reference, whereas U-L1 and U-L2a lie below the line and therefore exhibit underestimation. These differences can be explained by the mismatch in the calibration constants reported in Figure 4. The MFG calibration for U-S1 is larger than the reference calibration constant, which elevates the converted values, while those for U-L1 and U-L2a are smaller and thus depress the converted values. U-L2a shows the largest deviation, with a regression slope of approximately 0.592 attributable to coating damage. In Figure 6b, which shows the relative error of the MFG heat flux with respect to the reference, U-S1 and U-L1 display decreasing absolute error as the reference heat flux increases, followed by a gradual plateau. This behavior is similar to that observed for the new HFMs in Figure 5b. Quantitatively, U-S1 exhibits relative errors of approximately +3% to +14%, indicating the highest accuracy among the used HFMs. U-L1 ranges from approximately −25% to −12%, which is comparable to the new HFMs. In contrast, U-L2a shows the largest negative error, approaching −40% over the entire range, and its variation with heat flux differs from that of the other HFMs.

Summarizing the results of Figure 5 and Figure 6, the relative error of a used HFM with seven years in service and no visible coating damage did not differ markedly from that of the new HFMs. Counterintuitively, U-S1, which had about ten years in service and exhibited partial coating damage, shows the smallest absolute relative error. This finding was primarily attributed to U-S1 having been produced by a different manufacturer and calibrated using a different method, as reported in Table 1. In contrast, complete loss of the sensor coating led to very large measurement errors. Overall, these results showed that, when using the MFG calibration, the heat flux bias varied widely with sensor manufacturer, calibration method, service history, and coating condition. Therefore, ensuring repeatability and reproducibility through pre-use recalibration against the same secondary standard is as important as evaluating measurement accuracy.

To assess the effects of severe surface degradation and subsequent recoating on calibration feasibility and measurement reliability, Figure 7 presents U-L2a (coating completely removed) and U-L2b (recoated; ε = 0.9), in which heat flux values determined using the MFG calibration are compared with the reference (SEC) heat flux, and the corresponding relative errors are reported. In Figure 7a, a pronounced departure of the U-L2a slope from the reference line was observed. In contrast, the U-L2b slope was observed to approach the reference line, indicating that the converted heat flux became much closer to the reference. In Figure 7b, a large negative relative error of about −40% across the entire range was observed for U-L2a, while for U-L2b the error was reduced to an average of about −10%. These results indicate that recoating the sensing area can substantially reduce the relative error under the MFG calibration and that recalibration against a secondary standard can make damaged HFMs suitable for reuse. However, for HFMs with fully damaged coating, the response as a function of heat flux was not fully restored even after recoating.

Taken together, the results indicate that both the sensing-surface condition and the manufacturer-specific calibration methodology are key drivers of changes in the calibration constant. Coating degradation or damage lowers sensitivity and degrades linearity, whereas recoating partially restores these deviations toward the MFG level. This evidence sets the stage for the subsequent statistical analysis, which quantitatively interprets the systematic differences in calibration constants.

Figure 8 is a Bland–Altman plot that shows, for all HFMs examined, the difference between the MFG and SEC calibration constants plotted as MFG − SEC against their mean. The vertical axis represents the simple difference with SEC taken as the reference. Negative values indicate that the MFG calibration constant was smaller than the SEC calibration constant. The dotted lines show the mean difference of −1.58 kW/m^2^·mV and the limits of agreement (LOA) of −1.95 and −1.22 kW/m^2^·mV, computed from the new HFMs. By convention, ±1.96σ contains about 95% of the differences, so the dotted lines define the typical bias range for the new HFMs. First, with the secondary standard taken as a reference close to the true value, differences from the MFG calibration constants were examined. The smallest difference was observed for U-S1, and the next smallest negative differences were observed for U-L1 and U-L2b. By contrast, the new HFMs exhibited a larger average negative difference of approximately −1.58 kW/m^2^·mV, which implies that, under the assumption that the secondary standard is accurate, their bias may exceed that of used HFMs. In a complementary view reflecting common practice, new HFMs are often used without additional verification and are taken as an accuracy reference. Under this assumption, the new-HFM data were tightly clustered around the mean line, indicating low dispersion and high precision. On this reference, U-L1 and U-L2b were located near the upper LOA and would be expected to fall within acceptable bounds after minor recalibration, whereas U-S1 was located outside the LOA with a positive bias, indicating that the sign of the bias can change when the manufacturer and the calibration method differ. Accordingly, the use of high-precision new HFMs as an accuracy reference does not adequately account for systematic differences arising from different manufacturers and calibration methods. Finally, U-L2a was located at approximately −4.5 kW/m^2^·mV, well below the lower LOA, indicating loss of validity of the MFG calibration constant.

This study did not independently evaluate the absolute accuracy of the secondary-standard HFM, so a direct assessment of HFM measurement accuracy is constrained. The Bland–Altman bias and LOA reported here should therefore be regarded as a relative evaluation referenced to the secondary standard, and conclusions about absolute accuracy should be withheld. It was found that heat flux bias varied markedly with sensor manufacturer, calibration method, service history, and the condition of the sensing-area coating. Accordingly, pre-use recalibration against the same secondary standard is the most practical means of ensuring repeatability and reproducibility across experiments and laboratories. Furthermore, a secondary-standard calibration procedure using the ISO 5660-1 cone heater is practical in terms of instrument access, cost, and operational simplicity, and it suits routine quality control, interlaboratory comparisons, and performance verification after recoating or repair. Future work is needed to quantify the absolute accuracy of the secondary standard through cross-validation against a primary standard. It should also analyze sources of bias across manufacturers and calibration methods, standardize calibration models, and establish an absolute decision framework that encompasses measurement accuracy.

## 4. Conclusions

The practical utility of a calibration procedure referenced to a secondary standard, using an ISO 5660-1 cone heater as the radiant heat source, was verified. The calibration constants of new, used, and recoated HFMs were compared (six HFMs in total: three new, three used/recoated). The bias between MFG and SEC calibration constants was quantified, and practical implications for field application were derived.

The MFG calibration constants for the new HFMs were consistently smaller than those referenced to a secondary standard, and the converted heat flux was therefore systematically underestimated across the entire range (SEC > MFG by −17.9%, −14.5%, −15.8% for the three new units; converted heat flux ≈ 82–86% of reference). For used HFMs, when the coating was partially damaged or showed no visible damage, the relative error did not differ appreciably from that of the new HFMs (e.g., U-S1 ≈ +3% to +14%, U-L1 ≈ −25% to −12% over 3–20 kW/m^2^). When the coating was completely damaged, the largest negative error was observed (U-L2a ≈ −40% across the range), and recoating substantially reduced the average error (U-L2b ≈ −10% on average). In the Bland–Altman analysis, the new-HFM data clustered tightly around the mean line, indicating high precision, and the mean difference together with the limits of agreement defined the typical bias range. However, if new HFMs are treated as an accuracy reference without additional verification, this criterion cannot be applied directly to HFMs from different manufacturers or with different calibration methods, and instances of bias sign reversal were observed.

In this study, the absolute accuracy of the secondary-standard HFM was not independently evaluated. Despite this limitation, pre-use recalibration against the same secondary standard is a practical way to secure repeatability and reproducibility across experiments and laboratories. Secondary calibration using the ISO 5660-1 cone heater is well suited for field application due to its accessibility and operational simplicity at modest cost. Future work should cross-validate the secondary standard against a primary standard, identify sources of bias across manufacturers and calibration methods, and standardize calibration models to enable an accuracy framework with uncertainty reporting.

## Figures and Tables

**Figure 1 sensors-25-07406-f001:**
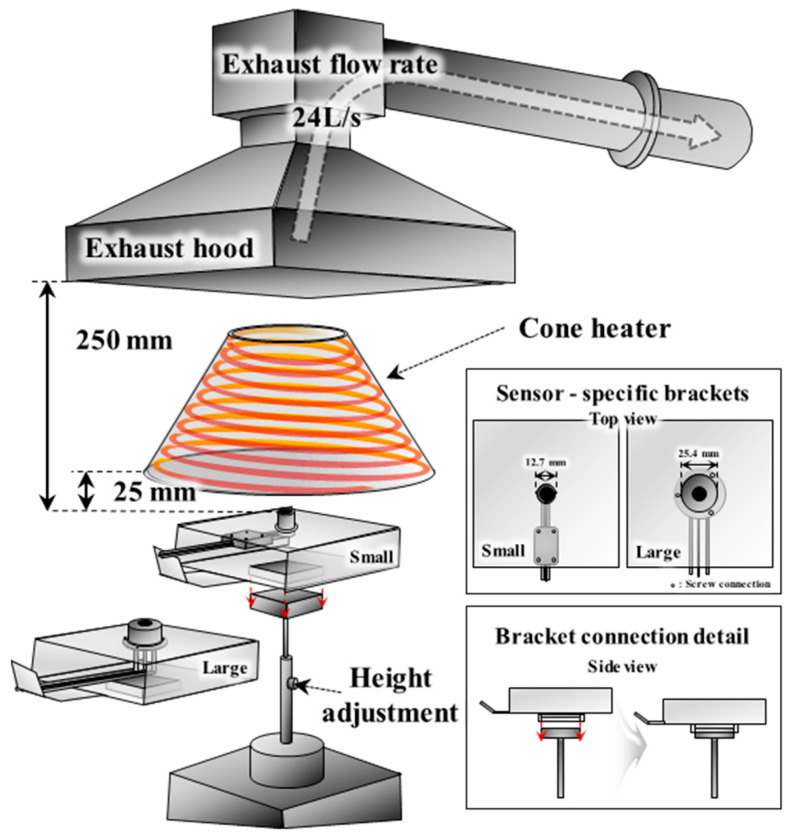
Schematic diagram of the HFMs calibration setup using a cone heater in accordance with ISO 5660-1.

**Figure 2 sensors-25-07406-f002:**
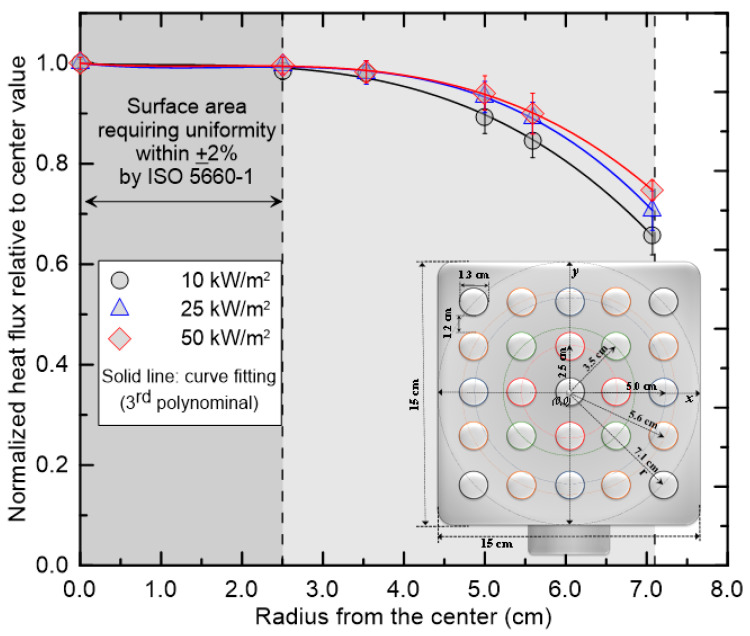
Radial distribution of normalized heat flux from a cone heater.

**Figure 3 sensors-25-07406-f003:**
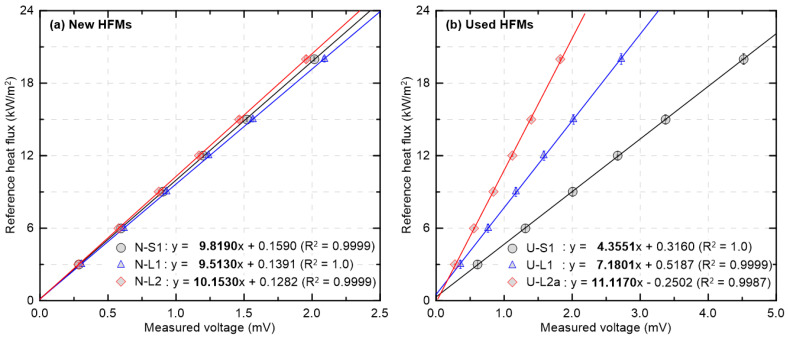
Linear calibration lines of new and used HFMs using a secondary-standard HFM.

**Figure 4 sensors-25-07406-f004:**
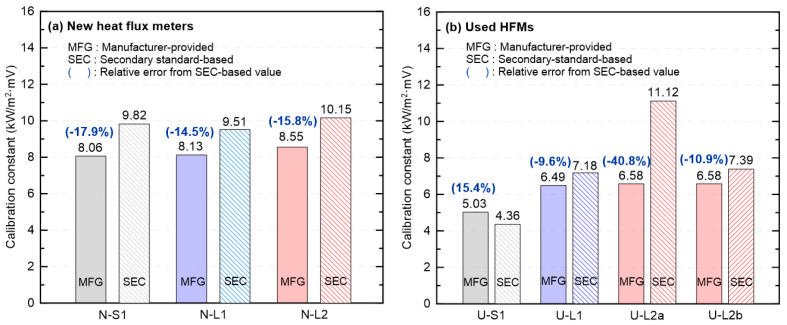
Comparison of calibration constants obtained from MFG and SEC data for new and used HFMs.

**Figure 5 sensors-25-07406-f005:**
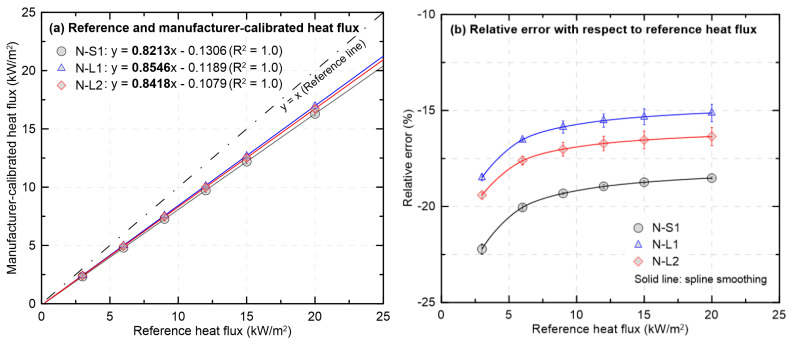
Deviation of the manufacturer-calibrated heat flux from the reference for the new HFMs.

**Figure 6 sensors-25-07406-f006:**
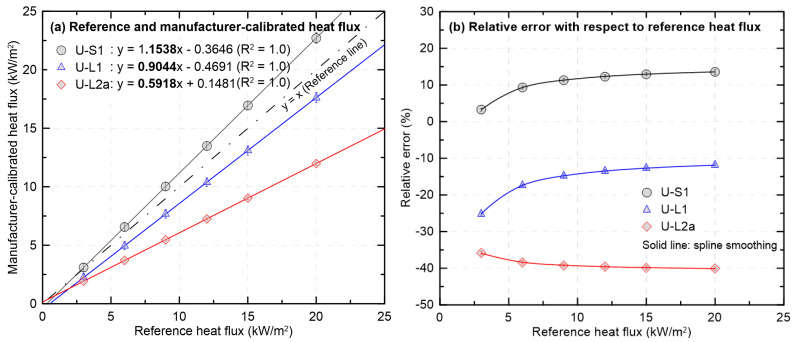
Deviation of the manufacturer-calibrated heat flux from the reference for the used HFMs.

**Figure 7 sensors-25-07406-f007:**
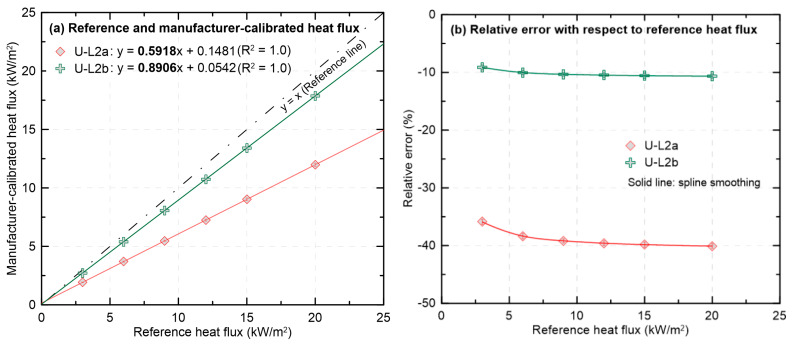
Deviation of the manufacturer-calibrated heat flux from the reference for the used HFM before (coating removed) and after recoating.

**Figure 8 sensors-25-07406-f008:**
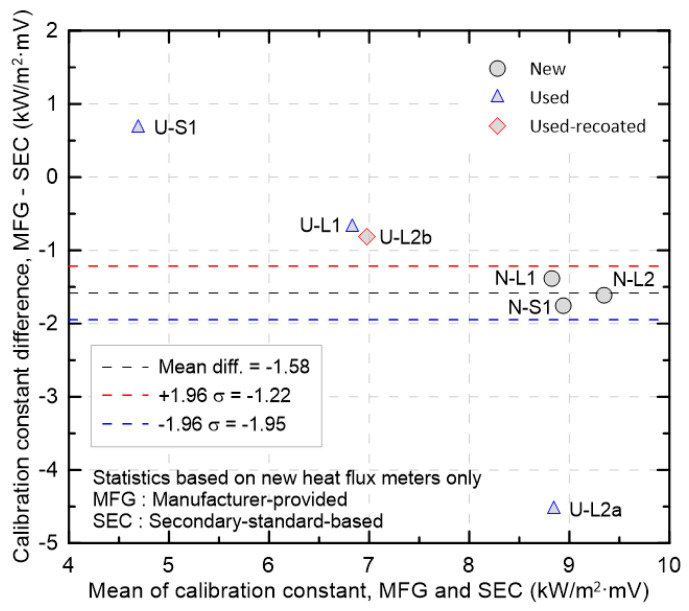
Bland-Altman analysis of calibration constants between MFG and SEC values, with limits of agreement from the new HFMs.

**Table 1 sensors-25-07406-t001:** Summary of HFMs used in the evaluation of calibration constant reliability.

Photo	SensorID	Condition	OuterDiameter (Inch)	SensorType	MaxHeat Flux (kW/m^2^)	YearsofUse	CalibrationConstant **(kW/m^2^·mV)	Manufacturer(Code)
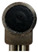	N-S1	New	0.5	Hybrid *	100	0	8.06	A
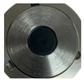	N-L1	New	1.0	Hybrid	200	0	8.13	A
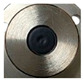	N-L2	New	1.0	Hybrid	200	0	8.55	A
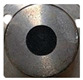	U-S1	Used—partial coating damage	0.5	Schmidt-Boelter	100	10	5.03	B
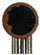	U-L1	Used—No visible coating damage	1.0	Hybrid	200	7	6.49	A
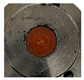	U-L2a	Used—coatingcompletely removed	1.0	Hybrid	200	7	6.58	A
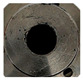	U-L2b	Used—recoated (ε = 0.9)

* Hybrid sensor combining Schmidt-Boelter and Gardon-type elements. ** Manufacturer-provided (MFG) calibration constant. All HFMs have a sensing diameter of 10 mm.

**Table 2 sensors-25-07406-t002:** Summary of the calibration constant and y-intercept for MFG and SEC values.

Sensor ID	Calibration Constant (kW/m^2^·mV)			y-Intercept (kW/m^2^)		
	Manufacturer- Provided (MFG)	Secondary-Standard-Based (SEC)		Manufacturer- Provided (MFG)	Secondary-Standard-Based (SEC)	
	Value	Value	Uncertainty	Value	Value	Uncertainty
N-S1	8.06	9.82	0.0904	0	0.159	0.1112
N-L1	8.13	9.51	0.0551	−7 × 10^−15^	0.139	0.0701
N-L2	8.55	10.15	0.0792	7 × 10^−15^	0.128	0.2360
U-S1	5.03	4.36	0.0261	1 × 10^−14^	0.316	0.0715
U-L1	6.49	7.18	0.0790	7 × 10^−15^	0.519	0.1294
U-L2a	6.58	11.12	0.3995	0	−0.250	0.4468
U-L2b		7.39	0.0886		−0.061	0.1471

## Data Availability

The data presented in this study are available on request from the corresponding author.

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
