# Peer review of "Evaluation of the Reliability of Calibration Constants in Heat Flux Meters Using an ISO 5660-1 Cone Heater"

_sensors, 2025, doi:10.3390/s25247406_

Round 1
Reviewer 1 Report
Comments and Suggestions for Authors
The paper systematically assesses the practicality of ISO 5660-1 conical heaters as secondary standard calibration sources. The comments are as follows:
1.The specific principles of different methods for calibrating heat flow meters need to be carefully analyzed, and it is recommended to supplement the schematic diagrams.
- The causes of calibration errors need to be carefully analyzed.
- The heat flux density error of the conical calorimeter is relatively large. Whether it can meet the requirements of the standard, it is suggested that the author supplement the analysis.
- It is recommended to construct a physical model for analysis to measure the quantitative relationship between voltage and heat flux, and take into account the influence of different factors.
- The influence of the test cooling water temperature (10-30°C) on the calibration constant needs to be analyzed.
- The calibration only covered the range of 3 to 20 kW/m² (secondary standard range), and the nonlinear response of the coating damage HFM under higher heat flux (such as 50 kW/m²) was not tested.
The English could be improved to more clearly express the research.
Author Response
Review 1
Comments 1:
The paper systematically assesses the practicality of ISO 5660-1 conical heaters as secondary standard calibration sources. The comments are as follows:
The specific principles of different methods for calibrating heat flow meters need to be carefully analyzed, and it is recommended to supplement the schematic diagrams.
Response 1:
We sincerely thank the reviewer for this valuable comment. As summarized by Pitts et al. [4], there is a broad spectrum of internationally recognized frameworks and practices for calibrating heat-flux meters (HFMs). These include the ISO standards for primary and secondary calibration (ISO 14934-2 and ISO 14934-3) and laboratory practices based on different radiant sources and geometries such as spherical blackbody furnaces, lamp/ellipsoidal systems, porous gas-fired panels, and blackbody targets with optical-pyrometry control. Representative guidance and practices (e.g., ISO 14934-2/-3, NT FIRE 050, NIST SP 250-65) are widely adopted by national laboratories and testing institutes worldwide. To avoid duplicating these well-documented principles, we direct readers to Pitts et al. [4] and related references for a comprehensive overview.
In the present study we focused on a practical secondary-standard calibration procedure that employs the ISO 5660-1 cone heater, which is readily available in many fire laboratories. Therefore, we limited our schematic to the detailed configuration of the cone-heater-based calibration setup as shown in Figure 1, while referring readers to the cited literature for broader calibration principles. We believe that the scope of Figure 1 is appropriate and sufficient for the objectives of this study, which aim to verify the practicality of the cone-heater-based secondary calibration. The comprehensive discussion of other calibration routes can be found in the cited references, and we kindly ask for the reviewer’s understanding regarding our decision to maintain this focused presentation in the current manuscript.
Comments 2:
The causes of calibration errors need to be carefully analyzed.
Response 2:
We appreciate the reviewer’s thoughtful comment. The physical causes of the calibration deviations between manufacturer-provided (MFG) and secondary-standard-based (SEC) constants are now clarified and discussed more explicitly in the revised manuscript. The additional paragraph inserted before the Bland–Altman analysis (Figure 8) summarizes the experimental findings from Figures 3–7 and emphasizes that both the sensing-surface condition and manufacturer-specific calibration methodology are the dominant factors affecting the calibration constant. Coating degradation or removal lowers sensitivity and linearity, while recoating partially restores the response toward the manufacturer value. These observations, together with the parameter differences summarized in Table 2, provide the physical basis for interpreting the systematic MFG–SEC bias in the subsequent statistical analysis.
Comments 3:
The heat flux density error of the conical calorimeter is relatively large. Whether it can meet the requirements of the standard, it is suggested that the author supplement the analysis.
Response 3:
We appreciate the reviewer’s valuable comment regarding the heat flux uniformity of the cone heater. As mentioned in the manuscript, the most critical aspect when using the cone heater as a radiation source for HFM calibration is the spatial distribution of heat flux density. To verify this, we measured the radial heat flux distribution, as presented in Figure 2. The results showed that the variation of heat flux was within ±2% up to a radial distance of 3.5 cm from the center, which satisfies the ISO requirement that allows a maximum 2% variation within a 2.5 cm radius.
Furthermore, considering that the sensing-surface diameter of the HFM is only 1 cm and the HFM was mounted precisely at the center of the specimen position, the limited non-uniformity of the cone heater does not cause a significant error in the calibration results. Therefore, the cone heater can be regarded as a practically reliable radiation source for secondary-standard-based calibration of HFMs.
Additionally, the temporal uniformity of the heat flux was confirmed under quasi-steady conditions. The 10-minute averaged variation in the measured heat flux was between 0.33% and 0.53%, consistent with previously reported stability data. These results collectively demonstrate that the heat flux stability and uniformity of the cone heater meet the requirements of the ISO 5660-1 standard and are suitable for practical calibration applications.
Comments 4:
It is recommended to construct a physical model for analysis to measure the quantitative relationship between voltage and heat flux, and take into account the influence of different factors.
Response 4:
We sincerely thank the reviewer for the helpful suggestion. In accordance with the comment, the revised manuscript explicitly clarifies the physical basis of the voltage–heat flux relationship, emphasizing that it is consistent with the thermoelectric response governed by radiative heat absorption at the sensing surface and subsequent heat conduction to the thermopile junctions. Specifically, the following statement has been added at the beginning of the Figure 3 description:
“The linear voltage–heat flux relationship observed here is consistent with a thermoelectric response governed by radiative heat absorption at the sensing surface and subsequent heat conduction to the thermopile junctions. Under these conditions, the slope represents an effective sensitivity determined by emissivity and cooling, whereas any intercept primarily reflects setup dependent offsets.”
This addition clarifies that the calibration constant (slope) can be physically interpreted as an effective sensitivity under given emissivity and cooling conditions, while the intercept mainly represents setup-dependent offsets.
Furthermore, the influence of key factors has already been addressed throughout the manuscript: coating degradation, removal, or recoating primarily affect the slope (sensitivity) and linearity (Figures 3–4 and 7), whereas coolant-temperature variation, background radiation/shielding, and electronics offset mainly influence the intercept, as discussed in Table 2 and the related text. The present study focuses on assessing the bias between manufacturer-provided (MFG) and secondary-standard-based (SEC) calibration constants, and this revision reinforces that the adopted linear calibration framework and its influencing factors are physically well-justified without introducing additional complex modeling.
Comments 5:
The influence of the test cooling water temperature (10-30°C) on the calibration constant needs to be analyzed.
Response 5:
We appreciate the reviewer’s insightful comment. In this study, the HFMs were supplied with cooling water at a flow rate of 0.7 L/min, and the temperature was maintained within the manufacturer-recommended range of 10–30 °C. The inlet temperature was 20.3 ± 0.6 °C, as stated in the manuscript. According to previous findings [11], variations in cooling-water temperature can cause shifts in the regression intercept (y-intercept) due to changes in the absolute heat flux through the sensor body, but they have no meaningful effect on the slope (calibration constant) that relates the output voltage to the absorbed heat flux.
Reference [11] (see its Figure 7) schematically illustrates this phenomenon: as the cooling-water temperature increases, the temperature difference between the sensing surface and the thermopile junctions decreases, leading to a leftward shift in the voltage–heat flux line (lower voltage at the same heat flux); conversely, when the cooling-water temperature decreases, the voltage increases (rightward shift). However, the slope remains constant, confirming that the calibration constant is essentially independent of coolant temperature within the recommended range.
In the present study, the inlet cooling-water temperature was tightly controlled within a narrow range, so any potential intercept variation was negligible. Consequently, the calibration constant itself was not affected. Since the objective of this work is to evaluate the bias between manufacturer-provided (MFG) and secondary-standard-based (SEC) calibration constants, which depends primarily on the slope, the influence of cooling-water temperature does not compromise the reliability of the calibration results.
[Please see the Figure of attachment]
Comments 6:
The calibration only covered the range of 3 to 20 kW/m² (secondary standard range), and the nonlinear response of the coating damage HFM under higher heat flux (such as 50 kW/m²) was not tested.
Response 6:
We fully understand and appreciate the reviewer’s concern. As shown in the manuscript, the HFM with completely damaged coating (U-L2a) exhibited the lowest linearity (R² = 0.9987), whereas the recoated U-L2b achieved an R² above 0.9999, showing nearly identical behavior to the other HFMs. This clearly confirms that the coating condition is the key factor governing linearity and sensitivity.
The calibration in this study was conducted within the secondary-standard range (3–20 kW/m²), consistent with the purpose of evaluating the practical reliability of SEC-based calibration using a cone heater. Nevertheless, it is widely recognized that water-cooled HFMs (e.g., Schmidt–Boelter type) exhibit excellent linearity over a much wider radiative range when the sensing-surface coating remains intact and cooling/alignment conditions are properly maintained. Based on this well-established characteristic and the consistently high linearity observed for the new and recoated sensors, it is reasonable to infer that the same linear correlation can be practically extrapolated to higher heat-flux levels, such as 50 kW/m², which are typical in cone-calorimeter testing.
However, for sensors with damaged or degraded coatings, the possibility of nonlinear response at high heat flux cannot be ruled out. Therefore, we recognize the importance of verifying this behavior directly in future work. In summary, the present study intentionally focuses on the reliability of SEC-range calibration, while acknowledging that the practical extrapolation beyond 20 kW/m² is valid only for intact, well-maintained HFMs, and that further testing at 25–50 kW/m² will be valuable to confirm the linear limit and ensure calibration safety.

Reviewer 2 Report
Comments and Suggestions for Authors
Dear authors,
The authors have carried out very interesting and useful research on various heat flux sensors using a cone calorimeter.
I have only one comment on the paper.
The authors use the term "uncertainty" only four times, while they frequently apply the outdated term "error". Moreover, it is completely unclear how the uncertainty calculations were performed. These calculations should be carried out in accordance with the Guide to the Expression of Uncertainty in Measurement (GUM), published by the Bureau International des Poids et Mesures (BIPM).
I believe the authors should add a section titled “Uncertainty Evaluation” and indicate uncertainty values on their graphs.
After these minor revisions, the paper can be accepted for publication.
Best regards, reviewer.
Author Response
Review 2
Comments 1:
The authors have carried out very interesting and useful research on various heat flux sensors using a cone calorimeter.
I have only one comment on the paper.
The authors use the term "uncertainty" only four times, while they frequently apply the outdated term "error". Moreover, it is completely unclear how the uncertainty calculations were performed. These calculations should be carried out in accordance with the Guide to the Expression of Uncertainty in Measurement (GUM), published by the Bureau International des Poids et Mesures (BIPM).
I believe the authors should add a section titled “Uncertainty Evaluation” and indicate uncertainty values on their graphs.
After these minor revisions, the paper can be accepted for publication.
Response 1:
We sincerely thank the reviewer for this constructive suggestion. We fully agree that the terminology of “uncertainty” and the approach to its evaluation should follow the framework of the Guide to the Expression of Uncertainty in Measurement (GUM) published by the BIPM. In the present study, however, a comprehensive GUM-based uncertainty analysis was not performed, because the primary objective was to assess the relative bias between calibration constants derived from manufacturer-provided (MFG) and secondary-standard-based (SEC) procedures rather than to establish traceable absolute uncertainties.
As stated in the manuscript,
“The listed uncertainties were obtained from linear-regression fits to the experimental data and represent two standard deviations (coverage factor k = 2). These experimentally derived uncertainties should not be interpreted as total calibration uncertainties [4].”
Therefore, the uncertainties shown in Table 2 correspond to statistical regression-derived spreads (Type A) rather than complete combined standard uncertainties as defined in the GUM. To avoid possible misunderstanding, we have clarified this definition in the revised text and ensured that the word uncertainty is used consistently throughout the manuscript in this intended sense.
Given that the study focuses on comparative calibration reliability rather than absolute metrology, a full GUM-compliant uncertainty budget is considered beyond the current scope. Nevertheless, we appreciate the reviewer’s valuable point, and we plan to incorporate a detailed Uncertainty Evaluation framework in future extended work dedicated to absolute calibration and traceability studies.

Reviewer 3 Report
Comments and Suggestions for Authors
Review of the Manuscript sensors-3967853-peer-review-v1
Responses to the following comments should be considered carefully:
- Ensure that all figure captions (especially Figures 3–8) are self-contained, clearly indicating the type of HFM (new/used, manufacturer A/B) and the measurement condition (e.g., recoated vs. uncoated). This will make the figures more understandable without referring back to the text.
- In Table 2, specify the number of calibration repetitions (n) used to determine uncertainties, and clarify whether the reported uncertainty represents standard deviation or expanded uncertainty (k = 2).
- Use consistent terminology throughout the manuscript for calibration constants (e.g., “MFG” vs. “manufacturer-provided”), and ensure all abbreviations are defined upon first appearance in the abstract or introduction.
- Verify that all ISO standards and NIST references follow MDPI reference style (e.g., include full titles and publication years consistently, and italicize standard names when required by Sensors formatting rules).
- Consider adding regression R² values directly on calibration plots (e.g., Figures 3 and 6) to quantitatively highlight linearity and improve the clarity of data reliability assessment.
“Minor Revision” status.
Author Response
Review 3
Comments 1:
Responses to the following comments should be considered carefully:
Ensure that all figure captions (especially Figures 3–8) are self-contained, clearly indicating the type of HFM (new/used, manufacturer A/B) and the measurement condition (e.g., recoated vs. uncoated). This will make the figures more understandable without referring back to the text.
Response 1:
We appreciate the reviewer’s thoughtful suggestion. To maintain concise figure captions while preserving clarity, all information regarding HFM types (new/used), manufacturers (A/B), coating conditions (uncoated/damaged/recoated), and sensor IDs has been clearly defined in Table 1. In addition, the Results section reiterates the specific HFM names (e.g., U-L2a, U-L2b) and test conditions at the point where each figure (especially Figures 3–8) is first discussed.
With this structure—cross-referencing Table 1 and providing brief in-text reminders—we believe readers can easily distinguish each HFM without expanding the figure captions, while preserving the manuscript’s readability and flow. We kindly ask for your understanding regarding this approach.
Comments 2:
In Table 2, specify the number of calibration repetitions (n) used to determine uncertainties, and clarify whether the reported uncertainty represents standard deviation or expanded uncertainty (k = 2).
Response 2:
We appreciate the reviewer’s helpful suggestion. Although the manuscript already states that three repetitions were performed for all conditions and that the relative standard deviation for repeatability is shown as vertical error bars, we have clarified the definition used in Table 2 for transparency. Specifically, the table note now reads:
“The listed uncertainties were obtained from linear-regression fits to the calibration data pooled over three repeated runs (n = 3) and are reported as two standard deviations (2σ) of the regression estimate. These values represent Type-A repeatability only and should not be interpreted as combined or expanded calibration uncertainties [4].”
This revision makes explicit both the number of repetitions (n = 3) and the meaning of the reported uncertainty (2σ, Type-A repeatability), addressing the reviewer’s request while keeping the manuscript’s scope unchanged.
Comments 3:
Use consistent terminology throughout the manuscript for calibration constants (e.g., “MFG” vs. “manufacturer-provided”), and ensure all abbreviations are defined upon first appearance in the abstract or introduction.
Response 3:
We appreciate the reviewer’s careful comment. In the revised manuscript, we have standardized all terminology related to calibration constants for consistency. Specifically, the expressions “MFG” and “manufacturer-provided” have been unified, and the corresponding term “SEC” is now consistently used for the secondary-standard-based calibration constant.
In addition, all abbreviations are now clearly defined at their first appearance in the Abstract and Introduction (e.g., “manufacturer-provided (MFG)” and “secondary-standard-based (SEC) calibration constants”). These editorial revisions ensure uniform terminology and improve readability throughout the manuscript.
Comments 4:
Verify that all ISO standards and NIST references follow MDPI reference style (e.g., include full titles and publication years consistently, and italicize standard names when required by Sensors formatting rules).
Response 4:
We appreciate the reviewer’s helpful comment. We have reviewed all ISO standards and NIST references cited in the manuscript and revised them to fully comply with the MDPI/Sensors reference style. The titles, publication years, and issuing organizations are now presented consistently, and the formatting (including italics for standard names where required) has been adjusted according to the journal’s guidelines.
These revisions ensure that all standards and institutional documents—such as ISO 5660-1, ISO 14934-2/-3, and NIST SP 250-65—follow a uniform citation format throughout the reference list and text.
Comments 5:
Consider adding regression R² values directly on calibration plots (e.g., Figures 3 and 6) to quantitatively highlight linearity and improve the clarity of data reliability assessment.
Response 5:
We appreciate the reviewer’s helpful suggestion. The calibration plots in Figure 3 already include the corresponding R² values to indicate linearity. In accordance with the reviewer’s comment, we have now added R² annotations to the additional calibration-related figures (Figures 5(a), 6(a), and 7(a)) to quantitatively highlight linearity and improve the clarity of data reliability assessment. These additions enhance the visual consistency and allow readers to easily evaluate the degree of linear correlation in each calibration plot.
Comments 6:
“Minor Revision” status.
Response 6:
We sincerely thank the reviewer for the positive evaluation and the recommendation for “Minor Revision.” All comments have been carefully considered and reflected in the revised manuscript.

Reviewer 4 Report
Comments and Suggestions for Authors
The intensity measurement probes of six different cone calorimeters are compared, including three calorimeters already in use.
Line 118: The NIST calibration method should be described in more detail. When a lamp is used as a radiation source, light with a different wavelength range than the thermal radiation is also present in the cone calorimeter. This topic should be explained in more detail.
L144: Due to the NIST comparison standard, very low irradiance intensities are used that are not used in normal measurement applications. It could be that the device no longer behaves linearly at higher intensities.
Line 163: What does “jig” mean?
Figures 4a and 4b: Same length of the y-axis
Due to the comparison standard, only irradiance levels up to 20 kW/m² are considered. According to EN 45 545-2, 25 kW/m² is required for railway materials and 50 kW/m² for rail transport products.
The conclusion should not only include what was done, but also the most important results as numerical values.
Author Response
Review 4
Comments 1:
The intensity measurement probes of six different cone calorimeters are compared, including three calorimeters already in use.
Line 118: The NIST calibration method should be described in more detail. When a lamp is used as a radiation source, light with a different wavelength range than the thermal radiation is also present in the cone calorimeter. This topic should be explained in more detail.
Response 1:
Thank you for highlighting two points: (i) clarification of the NIST lamp-based calibration method and (ii) the potential impact of spectral (wavelength-range) differences.
(i) NIST calibration method (clarified): The Methods section now includes a concise description of the NIST ellipsoidal-reflector system: “The NIST calibration source consists of a lamp and an ellipsoidal reflector and employs a 2000 W tungsten–halogen filament lamp. Its radiation passes through a 6.4 cm × 6.4 cm square aperture before reaching the sensor position, and the lamp output is controlled by a DC power supply, while the HFM signal is measured with a digital voltmeter [11].”
This addition provides the essential context (radiant source, optical geometry, measurement/control) while keeping the section succinct; detailed design and operation are in Ref. [11].
(ii) Spectral differences (interpretation and impact): As the reviewer correctly pointed out, the NIST lamp-based calibration source employs a 2000 W tungsten–halogen filament lamp with an ellipsoidal reflector, producing radiation with a relatively high color temperature that includes substantial visible and near-infrared components (≈ 0.4–2.5 µm). In contrast, the ISO 5660-1 cone heater, operating at around 900–1000 °C, emits predominantly in the mid- to far-infrared range (≈ 2–8 µm). Such spectral distribution differences can, in principle, cause variations in apparent sensitivity if the sensing-surface absorptance is not perfectly gray.
However, all heat-flux meters (HFMs) used in this study were coated with high-emissivity black surfaces designed to minimize wavelength dependence and provide nearly gray-body behavior. The experimental results also showed that variations in the calibration constant (slope) were primarily governed by coating condition (degradation or recoating), while offset shifts (intercepts) were associated with setup-dependent factors. This indicates that surface condition and setup configuration are the dominant influences rather than the radiative spectrum itself.
Furthermore, according to NIST Special Publication 1031 (“Round Robin Study of Total Heat Flux Gauge Calibration at Fire Laboratories”), interlaboratory calibrations conducted using different radiant sources (gas-fired panels, tungsten lamps, blackbody furnaces, etc.) showed agreement with manufacturer values within only a few percent (e.g., ± 2.4 % for Schmidt–Boelter and ± 5.5 % for Gardon gauges), confirming that spectral differences among sources exert only a minor effect on practical calibration consistency.
In conclusion, considering the purpose and scope of the present study—focused on establishing a practical and reproducible secondary-standard calibration using the ISO 5660-1 cone heater—the use of a secondary-standard HFM calibrated by a different radiant source is unlikely to introduce significant error for practical field applications.
Comments 2:
L144: Due to the NIST comparison standard, very low irradiance intensities are used that are not used in normal measurement applications. It could be that the device no longer behaves linearly at higher intensities.
Response 2:
We appreciate the reviewer’s concern. The calibration in this study was intentionally confined to the secondary-standard (SEC) range of 3–20 kW/m², within which all HFMs exhibited very high linearity (R² ≥ 0.9999 for new and recoated sensors). This range corresponds to the certified limit of the SEC reference and reflects the practical objective of the present work. It is well established that water-cooled HFMs maintain a highly linear response over wider radiative ranges when the coating is intact and cooling/alignment are properly controlled. However, nonlinear behavior may appear at higher fluxes if the coating is degraded or thermally stressed.
If a secondary-standard reference calibrated at higher irradiance levels becomes available in the future, we plan to conduct additional verification for the 25–50 kW/m² range using the same sensor configuration to further confirm linear behavior.
Comments 3:
Line 163: What does “jig” mean?
Response 3:
We appreciate the reviewer’s question. The term “jig” refers to a mechanical fixture or mounting device used to hold the heat-flux meter (HFM) securely in position during calibration. To avoid ambiguity, we have replaced the word “jig” with “mounting fixture” in the revised manuscript and clarified its role in the experimental setup description.
Comments 4:
Figures 4a and 4b: Same length of the y-axis
Response 4:
We appreciate the reviewer’s attention to detail. To improve visual consistency and facilitate direct comparison, the y-axis limits of Figures 4(a) and 4(b) have been adjusted to have the same maximum value. This ensures that the differences between the two manufacturers’ calibration biases can now be interpreted on a consistent scale.
Comments 5:
Due to the comparison standard, only irradiance levels up to 20 kW/m² are considered. According to EN 45 545-2, 25 kW/m² is required for railway materials and 50 kW/m² for rail transport products.
Response 5:
We agree that EN 45545-2 specifies higher external heat-flux conditions (e.g., 25 kW/m² for railway interior materials and 50 kW/m² for certain rail-transport products). The present study focuses on assessing the reliability of SEC-range (3–20 kW/m²) calibration using a cone heater, where intact HFMs showed excellent linearity (R² ≥ 0.9999). This supports the practical extrapolation of the calibration relationship to higher flux levels for well-maintained sensors under identical setups.
Nonetheless, such extrapolation should not be applied to damaged or degraded sensors. Once a higher-flux secondary-standard reference is established, we intend to further examine the 25–50 kW/m² range to verify the linear limit and confirm applicability to EN 45545-2 conditions.
Comments 6:
The conclusion should not only include what was done, but also the most important results as numerical values.
Response 6:
We appreciate the reviewer’s suggestion. The Conclusions section has been revised to include essential quantitative outcomes while preserving its original structure and brevity. Specifically, it now reports:
- SEC vs. MFG bias for new HFMs: SEC > MFG by −17.9%, −14.5%, −15.8% (converted heat flux ≈ 82–86% of the reference).
- Representative ranges for used HFMs: U-S1 ≈ +3% to +14%, U-L1 ≈ −25% to −12% over 3–20 kW/m².
- Worst-case damaged vs. recoated: U-L2a ≈ −40% (complete coating loss); U-L2b ≈ −10% after recoating.
To keep the Conclusions concise and avoid redundancy, detailed precision metrics (e.g., Bland–Altman statistics) remain in the Results section.
All revisions have been clearly highlighted in the revised manuscript for ease of review.

Round 2
Reviewer 1 Report
Comments and Suggestions for Authors
The author has carefully responded to the opinions, but the following problems still exist:
The paper lacks theoretical models (such as heat conduction or radiation transfer equations) to illustrate the heat transfer of the heat flow meter during the measurement process. It is suggested that the diagram of the heat transfer mechanism be supplemented.
Author Response
Review 1
Comments 1:
The author has carefully responded to the opinions, but the following problems still exist: The paper lacks theoretical models (such as heat conduction or radiation transfer equations) to illustrate the heat transfer of the heat flow meter during the measurement process. It is suggested that the diagram of the heat transfer mechanism be supplemented.
Response 1:
We sincerely appreciate the reviewer’s thoughtful comment.
The reviewer suggested adding theoretical heat-transfer models (such as heat-conduction or radiation-transfer equations) and a supplementary diagram illustrating the heat-transfer mechanism of the heat flux meter.
We fully understand the intent of this comment; however, after careful consideration, we believe that adding such theoretical models or diagrams would not align with the primary objective and scope of the present study. The fundamental principles governing total heat flux measurement—radiative heat absorption at the sensing surface and subsequent heat conduction within the sensor—are well-established, widely standardized, and extensively documented in the fire-science and heat-transfer literature. These principles are already referenced in the Introduction section through established sources (e.g., Refs. [1–3, 8–12]) and are not the focus of the present research.
The aim of this study is not to re-derive or revisit the general heat-transfer mechanisms of heat flux meters,
but rather to experimentally evaluate the reliability of manufacturer-provided calibration constants by comparing them with secondary-standard-based calibration results using an ISO 5660-1 cone heater. The core contribution of the work lies in analyzing calibration bias, service-history effects, coating degradation, and the statistical agreement through Bland–Altman evaluation—not in developing or explaining theoretical heat-transfer models. Since the manuscript focuses on calibration reliability under controlled thermal conditions rather than on modeling sensor physics, incorporating detailed theoretical equations or additional heat-transfer schematics would shift the scope away from its experimental purpose and may reduce clarity.
For these reasons, and to maintain consistency with the study’s intended scope, we respectfully believe that additional theoretical descriptions or diagrams are not necessary. We hope the reviewer will understand that the manuscript’s central objective is to assess calibration reliability through experimental comparison rather than to elaborate on heat-transfer theory, which is already well understood and broadly accepted in this field.
Reviewer 4 Report
Comments and Suggestions for Authors
The conclusions have become more informative thanks to the inclusion of numerical values and results. Some additional information has improved the content.
Author Response
Review 4
Comments 1:
The conclusions have become more informative thanks to the inclusion of numerical values and results. Some additional information has improved the content.
Response 1:
We sincerely appreciate the reviewer’s comment noting that the additional information has helped improve the content of the manuscript.
We understand this remark as a positive evaluation of the enhancements incorporated in the previous revision, and we regard it as a valuable perspective that may also be helpful for the future development of this research.
We are grateful for the reviewer’s thoughtful assessment and kind feedback.